# SAHSD: Enhancing Hate Speech Detection in LLM-Powered Web Applications via Sentiment Analysis and Few-Shot Learning

## Abstract

As large language models (LLMs) increasingly power web applications, including social networks, the challenge of moderating hate speech has become a critical concern for the Web. These LLM-powered applications, while offering near-human interaction capabilities, are vulnerable to harmful or biased content due to imperfect training data scraped from the Web. Current hate speech detection methods often struggle with limited annotated data, especially for real-time moderation on these platforms. This paper introduces Sentiment-Aided Hate Speech Detection (SAHSD), a novel approach designed to enhance hate speech detection specifically in LLM-powered web applications. By treating hate speech detection as a few-shot learning task, SAHSD utilizes sentiment analysis to refine pre-trained language models (LM) for improved accuracy in recognizing harmful content. SAHSD first employs publicly available sentiment datasets to train a sentiment analysis model, which is then fine-tuned by merging sentiment prompts with hate speech prompts, enabling efficient and accurate detection even with limited training samples. The effectiveness of SAHSD is demonstrated through experiments on widely used web-sourced datasets like SBIC and HateXplain. SAHSD achieves an exceptional F1-score of 0.99 with only 64 training samples and outperforms advanced techniques such as ToKen, MRP, and HARE, with significant improvements of 33% on SBIC and 95% on HateXplain. SAHSD surpasses GPT-4 in generalization performance across multiple datasets, showing an 8% improvement when trained on equal-sized samples. These results underscore SAHSD's potential to enhance content moderation in LLM-driven web platforms, contributing to a safer, more inclusive and accountable Web ecosystem.

## CCS Concepts

• **Security and privacy → Social aspects of security and privacy**.

## Keywords

Security, Machine Learning, Large Language Model, Hate Speech

## 1 Introduction

With the burgeoning advancement of LLMs (LLMs) like OpenAI's GPT series and others, the potential for these models to inadvertently generate or fail to identify hate speech in both inputs and outputs has become a grave concern. The propagation of hate speech through LLM-based web applications (i.e., ChatGPT) can magnify discrimination and prejudice against target groups, leading to significant psychological harm [29, 33]. Ferrara *et al.* [11] revealed that LLMs can sometimes perpetuate and amplify harmful biases in their training data. Similarly, Zhao *et al.* [33] reported that LLMs, if inappropriately moderated, could produce outputs that reinforce negative stereotypes. Given the pervasive use of LLMs in applications ranging from chatbots to content generation, robust hate speech detection within these systems is critical and plays an indispensable role in natural language processing (NLP).

**Motivation.** The motivation behind this research stems from the critical and urgent need to address the proliferation of hate speech within proliferating LLM-based web applications and its potential negative impacts. However, existing research still faces significant performance bottlenecks, especially in a few-shot context.

Earlier studies on hate speech detection focused on rule-based methods [17]. With the rapid advancement of deep learning, neural networks [13] and word embedding methods [10] have been increasingly employed. The rise of pre-trained LMs, exemplified by models like Bidirectional Encoder Representations from Transformers (BERT) [9], has become foundational in hate speech detection subsequent studies. Aluru *et al.* [3] demonstrated that BERT-based models outperform recurrent neural network-based models, while Kim *et al.* [16] introduced a masked rationale prediction prefinetuning method to enhance detection performance. Caselli *et al.* [5] re-trained BERT on the Reddit Abusive Language English (RAL-E) dataset to create HateBERT for detecting abusive language in social media. ToKen [2] introduced a method that relies on task decomposition and knowledge infusion in a few-shot context to enhance hate speech detection, but its performance is not particularly impressive. The urgent necessity for effective hate speech detection within LLM-based web applications, coupled with the inadequacies of current methodologies, provides the impetus for our research endeavors.

**Challenge.** The primary challenge lies in effectively detecting hate speech within LLMs, particularly in scenarios with limited annotated datasets. Acquiring comprehensive, real-world hate speech datasets for training hate speech detection models is notoriously challenging due to the sensitive nature of hate speech and ethical considerations. AlKhamissi *et al.* [2] proposed the Token method and tried to address this challenge by breaking a detection task into conditional generation subtasks and incorporating common sense to handle limited sample availability. Mehdad *et al.* [22] utilized n-grams, character-level, and text-level sentiment features and employed support vector machines (SVMs), which require relatively fewer labeled samples for hate speech detection. Zhou *et al.* [34] introduced a hate speech detection framework centered on sentiment knowledge sharing. While these methods mitigate the challenge of limited annotated datasets to some extent, they fail to identify hate speech effectively and overlook the semantic comprehension afforded by pre-trained LMs.

**Contribution of this study.** In this paper, we propose *Sentiment-Aided Hate Speech Detection* (SAHSD), a novel and effective approach to hate speech detection in a few-shot context for LLM-based web applications. SAHSD takes advantage of the typical display of negative sentiments in hate speech. It integrates sentiment analysis into prompt-tuning, thereby fine-tuning a pre-trained LM to more effectively identify potential hate speech. Moreover, SAHSD creates

an intermediate task that injects sentiment knowledge into prompts, guiding the pre-trained LM to prioritize sentiment analysis. As a result, SAHSD can significantly improve hate speech detection in scenarios with limited data samples.

The key contributions of this paper are summarized as follows.

- We propose SAHSD, a novel framework that effectively integrates sentiment analysis into prompts for hate speech detection in LLM-based web applications. This integration of sentiment knowledge effectively captures the subtle semantics of hate speech within sentences of natural language, and improves the detection rate of hate speech.
- A two-stage workflow is designed to transform a pre-trained light weight LM into a hate speech detector: It first constructs sentiment analysis as an intermediate task during pre-fine-tuning, followed by a dedicated fine-tuning phase focused on hate speech detection.
- Comprehensive experiments substantiate the efficacy of SAHSD. The performance of SAHSD with larger sample sizes is particularly remarkable, nearly perfect beyond 64 samples. On the HateXplain dataset, SAHSD exhibits an average detection rate of 95%, outperforming the closest competitor across all sample settings tested. This superiority extends consistently to out-of-distribution datasets and significantly outperforms many other SOTA solutions, including GPT-4.

The proposed SAHSD framework addresses a critical need for robust and scalable hate speech detection, which directly impacts content moderation, online harassment prevention, and the mitigation of harmful content propagation—key concerns within the security and privacy community. Hate speech detection is inherently challenging due to its subjective nature and potential annotator biases. Our approach goes beyond simply aligning with possibly biased labels. It offers semantic understanding that is independent of specific annotator perspectives. By focusing on the underlying sentiment patterns associated with hate speech, our approach introduces a more objective criterion for detection.

Moreover, the use of a few-shot learning framework allows SAHSD to generalize across different datasets and contexts, making it less reliant on the peculiarities or biases of any individual dataset. The model learns from broader sentiment cues and linguistic features, enabling it to perform robustly even when faced with variations in labeling criteria. Our experiments show that SAHSD outperforms existing methods not just by conforming to dataset labels, but by leveraging sentiment analysis to capture more intrinsic and contextually appropriate elements of hate speech.

## 2 Sentiment-Aided Hate Speech Detection

In this section, we first formulate the target problem and then provide an overall design of the proposed SAHSD framework, followed by a detailed explanation of its individual components.

### 2.1 Problem Formulation

Hate speech detection can be conceptualized as a text classification challenge. Consistent with conventional hate speech classification methods, we adopt a binary classification framework to classify texts into either a "hate speech" or "non-hate speech" category. Let $\mathcal{D} = \{(X^{(k)}, y^{(k)})\}_{k=1}^{N}$ be a hate speech dataset of $N$ text samples, where $X^{(k)}$ is the $k$-th sentence in the dataset $\mathcal{D}$ and $y^{(k)} \in Y = \{0, 1\}$ is a binary label indicating whether the sentence contains hate speech ($y^{(k)} = 1$) or not ($y^{(k)} = 0$). Hate speech detection aims to learn a function $f$ that maps a given sentence $X^{(k)}$ to its corresponding label $y^{(k)}$, i.e., $f(X^{(k)}) \rightarrow y^{(k)}$.

### 2.2 Overall Design

As depicted in Fig. 1, a schematic representation of SAHSD comprises two stages: sentiment injection and hate speech detection.

In the first stage of sentiment injection, SAHSD enhances a pre-trained LM through the use of prompts for an intermediate task (i.e., sentiment analysis). The objective is to formulate a prompt that injects sentiment knowledge. As shown on the left-hand side (LHS) of Fig. 1, this involves the integration of sentiment knowledge through prompts.

Within the SAHSD framework, we identify labels associated with the sentiment, i.e., positive, neutral, and negative, respectively. The tokens corresponding to these three label words in the model's vocabulary are denoted as:

$$\{e(Pos), e(Neu), e(Neg)\} \tag{1}$$

which are directly derived from the label words, and $e(\cdot)$ indicate am embedding function.

All above-mentioned tokens are optimized during training in the sentiment injection stage. The injected sentiment, stored as prompts, will be leveraged in the second stage of hate speech detection.

In the second stage of hate speech detection, we refine the pre-trained LM using prompts developed in the first stage. This phase involves employing a hate speech detection prompt, and the corresponding tokens to label words (i.e., Hate speech or Non-hate speech), denoted as $\{HATE_1, HATE_2\}$, and $\{e(Hate), e(NonHate)\}$, respectively. These are illustrated on the right-hand side (RHS) of Fig. 1. The core idea in the second stage is prompt-tuning a pre-trained LM with sentiment prediction as part of the inputs for hate speech detection.

### 2.3 Sentiment Injection

Obtaining a high-quality dataset for hate speech detection is a significant challenge, largely due to the presence of annotator biases in existing datasets. To address this, we develop a new prompt-tuning strategy that leverages sentiment analysis as an intermediate task, which shows great potential to enhance hate speech detection.

The proposed SAHSD approach adopts sentiment classification as an intermediate task, aiming to first predict the sentiment of each input sentence. In this approach, sentiment is categorized into three types: *positive*, *neutral*, and *negative*.

First, we define the sentence under hate speech detection $X = \{x_1, x_2, \cdots, x_N\}$, where $x_i$ is the $i$-th ($i = 1, 2, \cdots, N$) token in the sentence $X$, and $N$ is the total token count of the sentence. Before feeding the sentence $X$ to the pre-trained LM, $X$ merges with prompt templates and special tokens, transforming into a fixed token sequence:

$$\hat{X} = [CLS]X[SEP][SENTI][SEP], \tag{2}$$

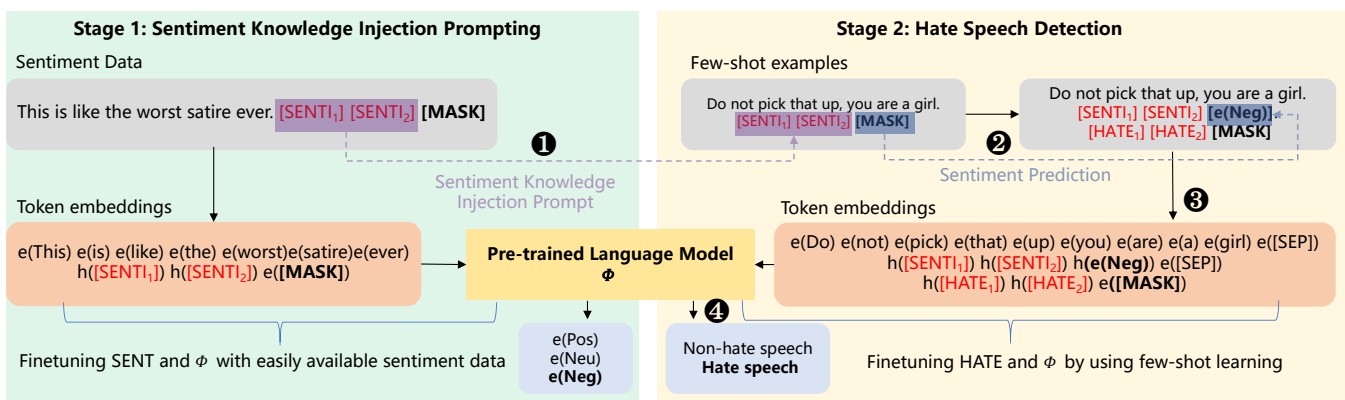

**Figure 1: An overview of the training process for SAHSD. In Stage 1 (the sub-figure on the left), we utilizes readily available sentiment data to finetune a sentiment analysis prompt "$SENTI_1, SENTI_2$", as well as the pre-trained LM $\phi$, resulting in a *Sentiment Knowledge Injection Prompt* (see ❶), which is then used in Stage 2 for sentiment prediction (see ❷). In Stage 2 (the sub-figure on the right), the hate speech detection template "$HATE_1, HATE_2$" is appended to the few-shot example, sentiment prompt, and sentiment prediction as input to the pre-trained LM for hate speech detection (see ❸). The hidden representations of the hate speech template and the pre-trained LM $\phi$ are finetuned to generate the correct detection result (see ❹). In both stages, $e(\cdot)$ represents the embedding of a token.**

where [CLS] and [SEP] are special tokens that stand for classification and separator, respectively.

$SENTI$ is the sentiment analysis prompt template defined as

$$SENTI = \{[SENTI_{0:m}], [MASK]\} \tag{3}$$

where $SENTI_i$ ($1 \leq i \leq m$) is the $i$-th token in the sentiment analysis template $SENTI$, and $m$ is the length of the template in terms of tokens.

The sentiment knowledge injection prompt is initialized from the meticulously crafted template $SENTI$. Then, the sequence $\hat{X}$ is mapped to a sequence of vectors by the pre-trained LM's input layer, denoted as $\{h_i \in \mathbb{R}^d\}$, where $0 \leq i \leq m$, and $m$ is the number of trainable word embeddings in the sentiment analysis prompt template $SENTI$, and $d$ is the dimension of the word embedding.

To inject sentiment into the prompts, we employ pseudo-tokens to design templates and use manual templates to initialize, as shown in (3). For instance, we can employ "It was" as the initial prompt element for the sentiment analysis stage, represented by the two tokens: $SENTI_1$ and $SENTI_2$. These pseudo-tokens correspond to unused tokens in the vocabulary of the pre-trained LM and are mapped to word embeddings by the pre-trained LM's input layer. Specifically, we map the template $[SENTI] = \{[SENTI_{0:m}], [MASK]\}$ to the following structure:

$$\{h_0, h_1, \cdots, h_m, h([MASK])\}, \tag{4}$$

where $h_i$ is the hidden representation of the token $SENTI_i$, and $h([MASK])$ is the hidden representation of the token MASK.

Since these hidden representations in (4) can be trained, the hate speech related sentiment can be incorporated as:

$$\hat{h}_{0:m} = \arg\min_{\theta} \mathcal{L}_s(\phi(\hat{X}; \theta), y_s), \tag{5}$$

where $\theta$ represents the collection of learnable parameters of the pre-trained LM, including the weights, bias, and the hidden representations in (4); $\mathcal{L}_s(\cdot, \cdot)$ is a cross-entropy loss of the sentiment

injection intermediate task (i.e., lower sentiment prediction accuracy indicate larger loss, and vise versa); $\phi(\cdot, \cdot)$ represents the pre-trained LM; $y_s$ is the label of the sentiment class.

This strategy takes advantage of the abundance of readily available sentiment analysis datasets. Specifically, we fine-tune a pre-trained LM by embedding sentiment knowledge into a learned continuous-style prompt during the adaptation process. This enriched prompt provides a robust foundation for the subsequent hate speech detection. By doing so, we effectively augment the model's understanding and capabilities, making it more adept at detecting hate speech. This method not only compensates for the limitations of available hate speech datasets but also taps into the nuanced relationship between sentiment analysis and hate speech, leading to more accurate and reliable hate speech detection.

## 2.4 Hate Speech Detection

Hate speech detection is defined as a binary text classification task, where the objective of the model is to determine whether a given input sentence is *hate speech* or *non-hate speech*. The hate speech detection model builds upon the sentiment-injection technique described in Section 2.3. Prior to hate speech detection, the detection model inherits the parameters acquired from the sentiment injection stage and uses the sentiment knowledge-injected prompts developed in that stage to predict the sentiment polarity of each potential hate speech.

The hate speech detection model is constructed based on the sentiment-injection technique described in Section 2.3. Given an input sequence $X$, the formatted input for hate speech detection is

$$\tilde{X} = [CLS]X[SEP][SENTI\_P][SEP][HATE][SEP], \tag{6}$$

where [SENTI_P] is the template containing the sentiment knowledge-injected prompt and the sentiment prediction. Meanwhile, [HATE] follows the template structure of the sentiment injection stage, tailored for the hate speech detection task. [SENTI_P] and [HATE]

are given by

$$[\text{SENTI\_P}] = \{h_0, h_1, \cdots, h_{m_1}, h([v_s])\};$$
$$[\text{HATE}] = \{h_0, h_1, \cdots, h_{m_2}, h([\text{MASK}])\},\qquad(7)$$

where $m_1$ and $m_2$ denote the numbers of trainable hidden representations in [SENTI_P] and [HATE], respectively; $h([v_s])$ represents the hidden representation of the token corresponding to the sentiment prediction verb for the original input sentence $X$.

As in the sentiment injection stage, the cross-entropy loss function is employed, leading to the final objective function:

$$\arg\min_{\theta} \mathcal{L}_h(f(\tilde{X}; \theta), y_h),\qquad(8)$$

where $\mathcal{L}_h(\cdot, \cdot)$ is a cross-entropy loss of hate speech detection; $y_h$ is the label word of the hate speech class.

## 3 Experiment

This section introduces the datasets and evaluation metrics used in our study. We also outline the comparative model methods and the experimental settings. Finally, we analyze the results of our experiments and discuss the generalizability of the method. Unless otherwise specified, we use the BERT-base-uncased model [9] as the pre-trained LM. This is because the BERT-base-uncased model has only 0.34 billion parameters (compared to the 13 billion parameters in the LLaMA2 model) and can achieve near-perfect performance in hate speech detection with just 64 training samples (see Table 1). This strategy allows us to fully leverage the language understanding capabilities of the backbone model with lower computational cost. The experiment setup is detailed in Appendix C.

### 3.1 Performance of Hate Speech Detection

In the proposed SAHSD framework, we fine-tune both the token embeddings and the pre-trained LM (BERT) to fully leverage the model's capabilities while integrating sentiment knowledge. This strategy strikes a balance between performance and computational efficiency, particularly in low-resource settings. Our SAHSD method achieves superb performance even with relatively few hate speech samples, evident for the near-perfect F1-score of 0.99 using only 64 samples (see the fifth column of Table 1).

While more extensive fine-tuning (e.g., fine-tuning with the entire training dataset of a hate speech task) could offer additional performance insights, the gain is likely to be marginal over the already achieved F1-score of 0.99. Moreover, we are particularly interested in scenarios where labeled hate speech data and computational resources are limited. This makes the SAHSD practical and scalable, even without the need for a large amount of labeled hate speech data and heavy resource investments.

Table 1 presents the performance of hate speech detection as measured by the F1-score of state-of-the-art methods, along with our proposed SAHSD framework, on the SBIC and HateXplain datasets. It is noted that both the BERT-H and BERT-M methods rely on a unique feature in the HateXplain dataset, specifically, the keywords that assist in determining whether a data point should be classified as hate speech. These two methods are not compatible with the SBIC dataset. Furthermore, the task decomposition strategy used by the ToKen method [2] is limited to binary classification, making it unsuitable for the HateXplain dataset, which includes

|  | Methods | Sample Size | | | | | | | |
|---|---|---|---|---|---|---|---|---|---|
|  |  | 16 | 32 | 64 | 128 | 256 | 512 | 1024 | Avg. |
| SBIC | BERT [9] | 0.59 | 0.61 | 0.71 | 0.73 | 0.77 | 0.80 | 0.83 | 0.72 |
|  | BART-P [18] | 0.45 | 0.53 | 0.56 | 0.60 | 0.64 | 0.70 | 0.73 | 0.60 |
|  | ToKen [2] | 0.59 | 0.63 | 0.70 | 0.70 | 0.70 | 0.72 | 0.73 | 0.68 |
|  | HARE [31] | 0.58 | 0.66 | 0.73 | 0.76 | 0.78 | 0.82 | 0.86 | 0.74 |
|  | Ours | **0.76** | **0.96** | **0.99** | **0.99** | **0.99** | **0.99** | **0.99** | **0.96** |
| HateXplain | BERT [9] | 0.28 | 0.33 | 0.41 | 0.48 | 0.52 | 0.57 | 0.60 | 0.45 |
|  | BART-P [18] | 0.26 | 0.34 | 0.43 | 0.52 | 0.58 | 0.60 | 0.62 | 0.48 |
|  | BERT-H [21] | 0.27 | 0.29 | 0.35 | 0.49 | 0.53 | 0.57 | 0.60 | 0.44 |
|  | BERT-M [16] | 0.20 | 0.23 | 0.24 | 0.35 | 0.55 | 0.55 | 0.59 | 0.39 |
|  | Ours | **0.74** | **0.92** | **0.98** | **0.98** | **0.98** | **0.98** | **0.98** | **0.94** |

**Table 1: Hate speech detection performance of the considered methods on the SBIC and HateXplain datasets. Each F1-score represents the average from ten random seeds.**

three label categories. To showcase the optimal performance of the HARE method [31], the experiments did not modify the details of the CoT prompts. Therefore, HARE was solely used for the binary classification task on the SBIC dataset.

As shown in Table 1, the proposed SAHSD technique consistently surpasses its counterparts, achieving the highest scores by a substantial margin on both the SBIC and HateXplain datasets. Overall, the SBIC dataset records higher scores than the HateXplain dataset. The HateXplain dataset, designed for three-class classification, is presumed to present a more complex challenge than the SBIC dataset. In the case of the SBIC dataset, ToKen demonstrates superior performance with a sample size of 16, while HARE prevails for sample sizes above 16. This indicates that the ToKen method has a distinct advantage in scenarios with extremely limited sample sizes. At the same time, the HARE method fully leverages the performance of large LMs and achieves a performance level that is just below our proposed method.

Regarding the HateXplain dataset, techniques focused on identifying critical segments of sentences—pre-labeled within the dataset itself (like BERT-H and BERT-M)—tend to underperform. This may be due to these methods requiring a large volume of samples to accurately identify key sentence segments indicative of hate speech. Notably, the proposed SAHSD approaches near-perfect performance once the sample size exceeds 64. On average, it achieves a relative improvement of **33%** compared to the second-best method on the SBIC dataset. Similarly, when tested on the HateXplain dataset, our method shows a relative improvement of **95%** over the next best method. These findings highlight the effectiveness of SAHSD, establishing it as a crucial intermediary task that significantly enhances the efficiency of hate speech detection.

### 3.2 Generalizability

Since the out-of-distribution datasets are binary-labeled, we evaluate the proposed SAHSD model, which was trained on the SBIC dataset, in a zero-shot manner on these two datasets without additional training. Fig. 2 illustrates the detection performance of SAHSD on these out-of-distribution datasets. Each point on the graph represents the average F1 score corresponding to a specific sample size. We conduct ten tests for each sample size and then calculate the average F1 score.

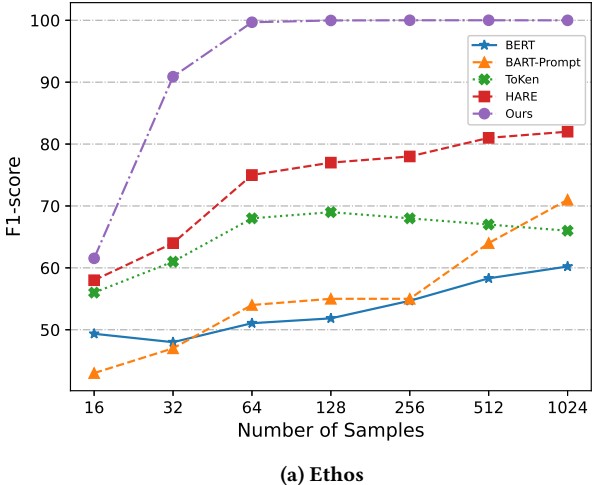

(a) Ethos

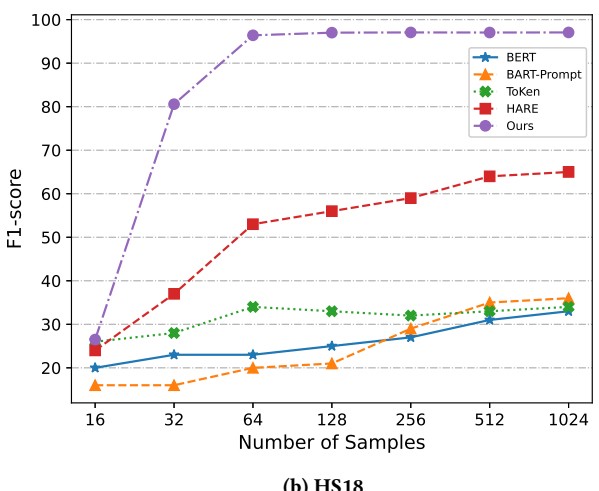

(b) HS18

**Figure 2: Zero-shot performance of out-of-distribution datasets HS18 and Ethos**

We see from Fig. 2 that SAHSD consistently outperforms the other alternative techniques on both datasets, approaching 100 after the number of samples reaches 64. Particularly on the HS18 dataset, the F1 scores of the other methods remain below 40 except for HARE, indicating their limited generalization capabilities. In contrast, SAHSD demonstrates commendable generalizability, especially in scenarios with limited sample sizes.

### 3.3 Performance on LLM-Generated Content

To measure the performance of the SAHSD framework on potential hate speech generated by LLMs, we evaluate the model in a zero-shot manner on the ToxiGen dataset, which was trained on the SBIC dataset with different training sample sizes without any additional training. Fig. 3 displays the results of our experiments, where each

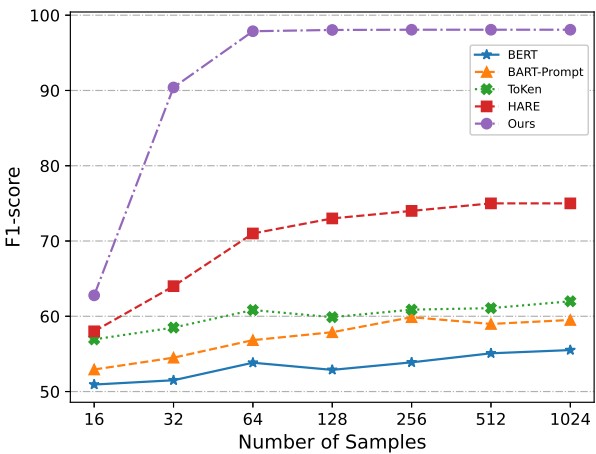

**Figure 3: The performance of the proposed method on the ToxiGen dataset in a zero-shot setting.**

result represents the average F1-score of the model trained with the corresponding number of training samples. For each model, we conduct ten tests with a certain number of training samples and took the average.

As shown in Fig. 3, the other methods slightly improve as the number of training samples increases. However, overall, they perform significantly worse than the proposed SAHSD method. Moreover, our SAHSD method demonstrates nearly perfect results when $N \geq 64$. It is evident that our method maintains robust detection for potential hate speech generated by LLMs.

### 3.4 Performance Comparison with LLMs

To compare the performance with LLMs in terms of generalization, we continue to conduct zero-shot comparisons. Our SAHSD method follows the settings mentioned in the generalizability section. For the experiments with the LLMs, we follow the CoT method outlined in the method proposed in [12]. Fig. 4 presents the experimental results on three generalization datasets. Since all experiments with the LLMs are conducted in a zero-shot scenario, the corresponding results are represented by a horizontal line in Fig. 4.

We see from Fig. 4 that among the LLMs, the GPT series, particularly GPT-4, exhibits the best performance. Our SAHSD method demonstrates comparable zero-shot performance to GPT-4 when trained on 32 samples. Moreover, as the number of training samples exceeds 32, our method achieve an average generalization performance that surpasses GPT-4 by **8%**. This indicates that SAHSD enables the detector, trained on a small amount of data, to outperform the current state-of-the-art LLMs in terms of generalization. These results highlight the superior performance of our method. Notably, the BERT-base-uncased model used in our approach, with only 0.34 billion parameters, is dramatically smaller than the smallest of the LLMs compared, such as Vicuna-13b with 13 billion parameters. This explains why our method performs slightly worse than these LLMs when using just 16 samples. However, obtaining more than

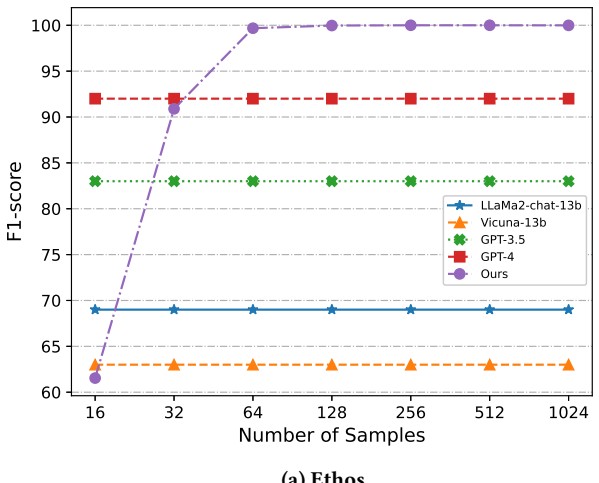

(a) Ethos

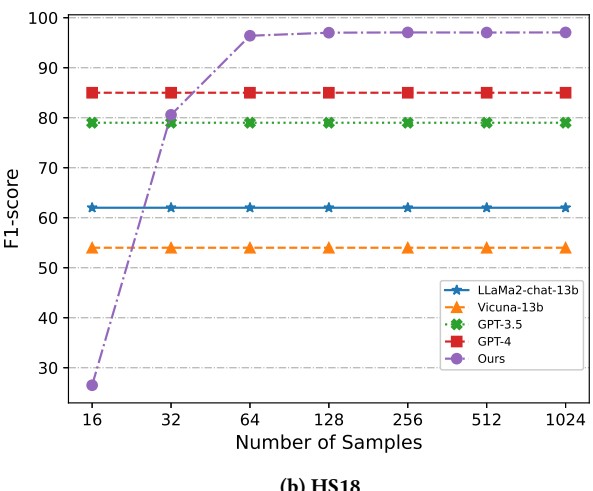

(b) HS18

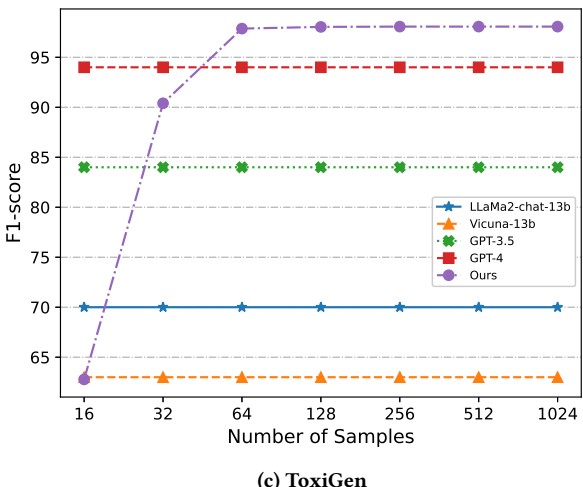

(c) ToxiGen

**Figure 4: Zero-shot performance comparison with LLMs**

| No. | Conf. | Sample Size | | | | | | | |
|---|---|---|---|---|---|---|---|---|---|
| | | 16 | 32 | 64 | 128 | 256 | 512 | 1024 | Avg. |
| 0 | default | **0.76** | **0.96** | **0.99** | **0.99** | **0.99** | **0.99** | **0.99** | **0.96** |
| 1 | w/o SAHSD | 0.63 | 0.68 | 0.73 | 0.77 | 0.81 | 0.83 | 0.83 | 0.75 |
| 2 | w/o Prompts | 0.62 | 0.69 | 0.73 | 0.79 | 0.82 | 0.84 | 0.84 | 0.76 |
| 3 | w/o PM | 0.65 | 0.69 | 0.73 | 0.78 | 0.83 | 0.83 | 0.84 | 0.76 |

**Table 2: Ablation Study. (1) w/o SAHSD: Instead of executing pre-finetuning intermediate tasks, we solely rely on prompt-tuning for hate speech detection; (2) w/o Prompts: We omit the sentiment knowledge-injected prompts from the first stage during second-stage training, employing only the prompts and the pre-finetuned model specifically crafted for hate speech detection; (3) w/o PM: We exclude the pre-finetuned model from the first stage in the second-stage training. We use the pre-trained model, incorporating sentiment prediction obtained from the pre-finetuned model, for detecting hate speech.**

32 hate speech samples from web-based social networks is straightforward, ensuring that our method delivers excellent performance in real-world scenarios.

## 4 Empirical Analysis

In this section, we conduct an ablation study of the proposed SAHSD framework. Additionally, we analyze the impact of initialization and model scale on SAHSD.

### 4.1 Ablation Study

Table 2 presents the results of our ablation study on the SBIC dataset. The first group demonstrates the outcomes when SAHSD is not used as an intermediate task. This results in a **21%** reduction in average hate speech detection performance relative to the default setting, underscoring the importance of SAHSD in the process. The second group showcases the results without incorporating sentiment knowledge-injected prompts for hate speech detection. This leads to a **20%** drop in average performance compared to the default setting, but a **1%** improvement over the first group. This suggests that the pre-finetuned model can integrate some external sentiment knowledge that benefits the hate speech detection task.

The third group represents the outcomes without using the pre-finetuned model as the backbone for hate speech detection. Instead, it employs sentiment prediction derived from the pre-finetuned model in lieu of sentiment knowledge-injected prompts. Since these prompts depend on the model's vocabulary, they are not applied here. The results show a **1%** improvement compared to the first group and a **0.6%** enhancement relative to the second group. This indicates that sentiment predictions from the pre-finetuned model can aid hate speech detection, with their influence outweighing that of the pre-finetuned model itself.

## 4.2 Correlation between Sentiment and Hate Speech

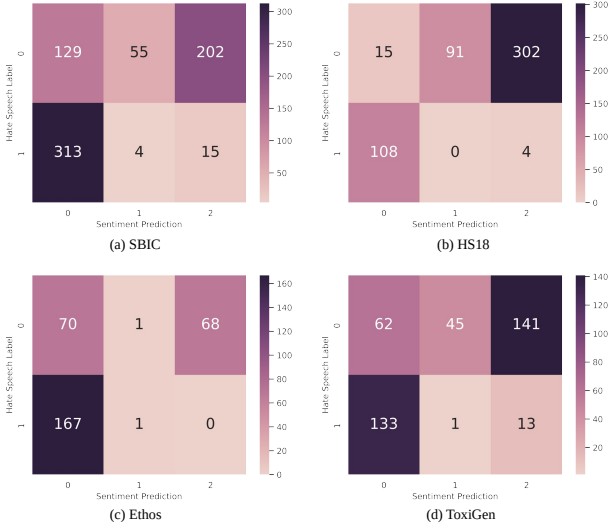

**Figure 5: Correlation Heatmap between sentiment and hate speech on datasets SBIC, HS18, Ethos and ToxiGen**

Next, we validate the rationale behind the proposed SAHSD method, demonstrating why embedding sentiment analysis into the hate speech task outperforms the existing approaches. To explore the correlation between sentiment and hate speech, we employ the most effective sentiment classifier from the first-stage training to conduct sentiment analysis on the SBIC, HS18, Ethos, and ToxiGen datasets. We specifically filtered out samples with a confidence level exceeding 0.99 and subsequently generated a correlation heatmap between the predicted sentiment labels and the hate speech labels. As shown in Fig. 5, a strong correlation is between negative sentiment and the presence of hate speech in sentences. Conversely, positive sentiment tends to correlate with the absence of hate speech. These observations provide empirical support for the efficacy of our approach in identifying hate speech based on sentiment analysis.

## 4.3 Impact of Prompt Initialization

Tables 3 and 4 present the hate speech detection performance achieved using different initialization templates for the first-stage sentiment analysis and the second-stage hate speech detection tasks, respectively, on the SBIC dataset. Our approach involves employing manually designed templates for initializing prompt tokens. An analysis of the two tables reveals that changing the initialization templates impacts performance, particularly with smaller training sample sizes. For example, with a training sample size of 16, Template-1 yields optimal results in the first stage, while Template-3 is most effective in the second stage. As the training sample size increases to 32, the default initialization template performs better. With further increases in training sample size, the performances of

| Prompt | Sample Size | | | | | | |
|---|---|---|---|---|---|---|---|
| Initialization | 16 | 32 | 64 | 128 | 256 | 512 | 1024 |
| default | 0.76 | **0.96** | **0.99** | **0.99** | **0.99** | **0.99** | **0.99** |
| Template-1 | **0.78** | 0.93 | 0.99 | 0.99 | 0.99 | 0.99 | 0.99 |
| Template-2 | 0.77 | 0.94 | 0.99 | 0.99 | 0.99 | 0.99 | 0.99 |

**Table 3: Impact of different prompt initialization templates on the sentiment analyses performance of the proposed method on the SBIC dataset. Under the default setting, the initialization template used is *"It was"*. Template-1 refers to *"The sentiment of this statement is"*, and Template-2 corresponds to *"The emotion of this sentence is"*.**

| Prompt | Sample Size | | | | | | |
|---|---|---|---|---|---|---|---|
| Initialization | 16 | 32 | 64 | 128 | 256 | 512 | 1024 |
| default | 0.76 | **0.96** | **0.99** | 0.99 | **0.99** | **0.99** | **0.99** |
| Template-3 | **0.82** | 0.94 | 0.99 | 0.99 | 0.99 | 0.99 | 0.99 |
| Template-4 | 0.77 | 0.93 | 0.99 | **0.99** | 0.99 | 0.99 | 0.99 |

**Table 4: Impact of different prompt initialization templates on the hate speech detection performance of the proposed method on the SBIC dataset. Under the default setting, the initialization template used is *"Offensive towards a group?"*. Template-3 refers to *"Verbal abuse directed at a group?"*, and Template-4 corresponds to *"Hate speech?"*.**

| Prompt | Sample Size | | | | | | |
|---|---|---|---|---|---|---|---|
| method | 16 | 32 | 64 | 128 | 256 | 512 | 1024 |
| default | **0.76** | **0.96** | **0.99** | **0.99** | **0.99** | **0.99** | **0.99** |
| PET | 0.44 | 0.52 | 0.63 | 0.65 | 0.66 | 0.66 | 0.66 |
| P-tuning v2 | 0.54 | 0.62 | 0.68 | 0.71 | 0.72 | 0.73 | 0.74 |

**Table 5: Impact of different prompt-tuning methods on the performance of the proposed method under the SBIC Dataset.**

different templates tend to converge. The influence of the initialization template diminishes as the sample size grows, allowing for more effective training of the prompt tokens.

## 4.4 Impact of Prompt Tuning

Table 5 showcases the hate speech detection performance achieved using various prompt tuning methods within our proposed method on the SBIC dataset. The Pattern-Exploiting Training (PET) method, serving as a foundational prompt learning approach in this experiment, involves pattern learning first for sentiment analysis and then for hate speech detection. In contrast, the P-tuning v2 method employs multi-layer prefixes for prompt learning, following a two-stage prompt learning process similar to PET. According to the table's results, the PET method exhibits the lowest performance, suggesting its limited capability to retain sentiment knowledge despite undergoing the first-stage sentiment analysis task. On the other hand, P-tuning v2 outperforms PET due to its greater number of task-specific parameters, which allows for better retention and optimization of external sentiment knowledge. Our proposed

| HS detection | Sample Size | | | | | | |
|---|---|---|---|---|---|---|---|
| Sample Size | 16 | 32 | 64 | 128 | 256 | 512 | 1024 |
| 32 | 0.60 | 0.68 | 0.74 | 0.88 | 0.92 | 0.95 | 0.95 |
| 64 | 0.59 | 0.65 | 0.73 | 0.87 | 0.93 | 0.99 | 0.99 |

**Table 6: Impact of different sentiment training sample sizes on the performance of the proposed method, under the SBIC dataset.**

| Sentiment Analyse Model | Sample Size | | | | | | |
|---|---|---|---|---|---|---|---|
| | 16 | 32 | 64 | 128 | 256 | 512 | 1024 |
| default | **0.76** | **0.96** | **0.99** | **0.99** | **0.99** | **0.99** | **0.99** |
| distilbert(setting 1) | 0.59 | 0.65 | 0.69 | 0.71 | 0.73 | 0.74 | 0.74 |
| distilbert(setting 2) | 0.69 | 0.74 | 0.81 | 0.84 | 0.85 | 0.85 | 0.86 |
| twitter-roberta(setting 1) | 0.58 | 0.64 | 0.69 | 0.70 | 0.73 | 0.73 | 0.74 |
| twitter-roberta(setting 2) | 0.69 | 0.73 | 0.79 | 0.83 | 0.83 | 0.84 | 0.84 |

**Table 7: Impact of different sentiment analysis models on the performance of the proposed method, under the SBIC Dataset.**

| Backbone Model | Model Size | Sample Size | | | | | | |
|---|---|---|---|---|---|---|---|---|
| | | 16 | 32 | 64 | 128 | 256 | 512 | 1024 |
| BERT-base | 110 M | 0.76 | 0.96 | 0.99 | 0.99 | 0.99 | 0.99 | 0.99 |
| BERT-large | 340 M | **0.78** | **0.96** | **0.99** | **0.99** | 0.99 | 0.99 | 0.99 |

**Table 8: Impact of backbone model size on the hate speech detection of the proposed method under the SBIC Dataset.**

method surpasses both, indicating the effectiveness of our specific prompt learning approach. This approach not only successfully injects sufficient sentiment knowledge into the prompts during the first stage but also efficiently utilizes this knowledge in the second stage. Such findings emphatically demonstrate the superiority of our method in hate speech detection performance.

### 4.5 Impact of Sentiment Training Samples

Table 6 displays the hate speech detection performance achieved using various sentiment training sample sizes within our proposed method on the SBIC dataset. The table reveals that when the training samples for hate speech detection are fixed at 32 and 64, there is a notable improvement in experimental results with an increase in the training samples for the first-stage sentiment analysis. Notably, when the training sample size reaches 512, the experimental results align closely with those obtained under the default settings. This suggests that the performance of hate speech detection is considerably enhanced when a sufficient number of training samples for sentiment analysis are provided.

### 4.6 Impact of Sentiment Analyse Model

Table 7 presents the hate speech detection performance achieved using different sentiment analysis models within our proposed method on the SBIC dataset. We select two state-of-the-art sentiment analysis models, distilbert-base-multilingual-cased-sentiments-student and twitter-roberta-base-sentiment-latest, to replace the model obtained from the first-stage sentiment analysis, thereby investigating their impact. 'Setting 1' involves using the sentiment analyzer directly in the second stage of hate speech detection for sentiment inference, with the results then being used to complete this stage. 'Setting 2' involves using the sentiment analyzer for sentiment inference in the second stage and mapping the sentiment results to sentiment knowledge-injection prompt tokens, which were trained in the default method, to complete the second stage.

The results in the table show that the performances of the two sentiment analysis models are comparable under the same setting. However, for the same model, Setting 2 consistently outperforms Setting 1. This suggests that while effective sentiment inference in Setting 1 aids the second stage to some extent, our method of injecting sentiment knowledge into the prompt tokens is more efficient in achieving the objectives of the second stage. Compared to Setting 2, the superiority of the default setting is most likely attributed to the unique representation of our trained sentiment prompt knowledge tokens, which are specifically tailored for sentiment knowledge injection after training and may not completely align with the representations used in other sentiment analysis models.

### 4.7 Impact of Model Size

Table 8 displays the F1-scores for hate speech detection achieved using BERT-base-uncased and BERT-large-uncased as the backbone models in our proposed method on the SBIC dataset. We observe that at sample sizes $N = 16$ and 32, the F1-score with BERT-large-uncased shows an improvement of **1.9**% and **0.6**% respectively, compared to BERT-base-uncased. However, for larger sample sizes ($N > 64$), the detection performance of the two models converges and becomes comparable. This suggests that increasing the model size improves results when dealing with very small sample sizes, but the impact diminishes with larger sample sizes. Based on these findings, we can infer that switching to a larger LM may not significantly enhance performance in cases of ample data. However, it is important to consider that while using a larger LM might not drastically boost detection performance, it could lead to substantial reductions in inference time and deployment costs.

### 5 Conclusion

In response to growing concerns about hate speech in LLM-based web applications, we propose SAHSD, a framework designed to enhance hate speech detection. LLMs risk spreading biased or inappropriate content due to unfiltered training data or user interactions. SAHSD tackles this with a two-stage training process, starting with sentiment analysis and then fine-tuning using both sentiment and hate speech prompts. This approach improves detection and adapts to various scenarios. Our experiments show that SAHSD achieves near-perfect F1-scores with minimal data and excels in zero-shot generalization, outperforming benchmarks. Future work will extend SAHSD to large-scale labeling of social media data.

# Appendix A: Notation and Definition

Table 9 summarizes the notation used in this paper.

**Table 9: Notation and definition**

| Notation | Definition |
|---|---|
| $X$ | A sentence under hate speech detection |
| $y$ | Hate speech label |
| $Y$ | The collection of hate speech labels, |
| $f(\cdot)$ | Hate speech detection function |
| $SENTI$ | A sentiment analysis template |
| $SENTI_i$ | The $i$-th token in a sentiment analysis template |
| $HATE$ | A hate speech detection template |
| $HATE_i$ | The $i$-th token in a hate speech detection template |
| $e(\cdot)$ | The embedding function |
| $h_i$ | The hidden representation of the $i$-th token |
| $\hat{h}_i$ | The optimized $h_i$ |
| $\phi(\cdot)$ | The pre-trained LM |
| $\mathcal{L}_s(\cdot, \cdot)$ | The loss function of sentiment analysis |
| $\mathcal{L}_h(\cdot, \cdot)$ | The loss function of hate speech detection |
| $F1_i$ | The F1 score of the $i$-th class |

# Appendix B: Related Work

In this section, we present a brief overview of related research work in the realm of hate speech detection. We start by discussing recent progress in hate speech detection, particularly focusing on advancements made in few-shot learning scenarios. Following this, we explore studies that utilize pre-finetuning as an intermediary step to enhance hate speech detection models. Finally, we assess the effectiveness of prompting techniques within LLMs and their implications for hate speech detection systems.

## B.1 Few-shot Hate Speech Detection

Gathering a large, high-quality labeled dataset of hate speech is challenging, primarily because genuine hate speech instances are relatively rare, and labeling is labor-intensive. This issue complicates the task of sampling online content (e.g., from social media) that contains hate speech without relying on specific keywords [25]. Additionally, hate speech often manifests subtly, without explicit profanity or obscenity, leaving no proper keywords for detection, which further complicates large-scale data collection [14].

Several studies in few-shot hate speech detection have focused on zero-shot or few-shot cross-lingual transfers. These methods use data-rich source languages to assist in detecting hate speech in target languages with less data [24]. For example, ToKen [2] introduced a method that relies on task decomposition and knowledge infusion, utilizing the BART pre-training model in a few-shot context to enhance hate speech detection, but its performance is not particularly impressive.

Mathew et al. [21] leverages the annotated labels in the HateXplain dataset to discern which segments of the posts should be emphasized. It integrates an attention supervision mechanism into the BERT model targeting those segments. Kim et al. [16] introduce the Masked Rationale Prediction (MRP) intermediate task before fine-tuning for hate speech detection based on method HateXplain [21]. However, these methods require a significant amount

of manual labor to annotate key sentence segments in the dataset, and their performance is not particularly impressive. HARE [31] is built upon the reasoning capability of LLMs, which are fine-tuned using chain-of-thought (CoT) prompts to train the model in identifying text categorized as hate speech. This method utilizes the powerful language capabilities of LLMs for multi-turn analysis of sentences to detect hate speech.

In contrast, our work delves into the effectiveness of a pre-trained-LM-based fine-tuning method combined with sentiment analysis. We explore how this approach can significantly improve hate speech detection, even with limited data availability.

## B.2 Pre-finetuning on an Intermediate Task

Pre-finetuning pre-trained models for subsequent tasks has recently gained traction [15]. This strategy involves training models to orient them before they engage with the target task. As explained by the Muppet method [1], pre-finetuning allows models to familiarize themselves with patterns relevant to the finetuning task, subsequently reducing the tuning duration, accelerating convergence, and enhancing task performance. Another approach proposed by [16] introduced an intermediate pre-finetuning task of masked rationale prediction, improving hate speech detection.

Recent studies have also utilized sentence sentiment features to refine hate speech detection. Malmasi et al. [20] conducted effective experiments demonstrating the fruitful application of n-gram and sentiment features in hate speech detection. Rodriguez et al. [26] compiled a hate speech dataset from Facebook and introduced a comprehensive suite of sentiment features closely linked to sentences—including negative sentiment words and symbols—to identify hate speech. Del et al. [8] used word sentiment values as the primary criterion for discerning hate speech within sentences. Furthermore, Zhou et al. [34] presented a hate speech detection framework based on sentiment knowledge sharing, which leverages external sentiment data sources and assimilates sentiment traits inherent to the target sentence. These studies collectively highlight the critical role of sentiment features in hate speech detection.

In line with insights from the Muppet method [1], a strong correlation between intermediate and target tasks can optimize pre-finetuning outcomes. Accordingly, we have developed a two-stage fine-tuning procedure for a pre-trained LM, designating sentiment analysis as an intermediate pre-finetuning task. This approach guides model learning using external sentiment insights and infuses the prompt with sentiment knowledge.

## B.3 Prompting for LLMs

Prompt tuning for LLMs evolved from the few-shot learning capabilities demonstrated by GPT-3 [4]. This method uses handcrafted prompts (contextual learning) to achieve remarkable few-shot performance. Pattern-Exploiting Training (PET) [28] converts NLP tasks into cloze-style questions. However, due to the continuous nature of neural network gradient optimization algorithms, discrete prompts are often sub-optimal. P-tuning [19] was developed to address this issue by employing an LSTM to encode the prompt as a continuous, trainable variable. This strategy requires optimizing network parameters external to the LLM, which can lead to complex label spaces. In contrast, method DART [32] introduced

the integration of word input layer parameters within the LLM (as unused tokens). These tokens are treated as pseudo tokens for prompts and optimized alongside the model during the gradient descent process.

Building on this continuous prompt-tuning technique, we propose a two-stage training scheme. This approach aims to enhance learning efficacy in few-shot scenarios. Additionally, we incorporate sentiment knowledge into the prompts during intermediate tasks. The resulting augmented prompt is used in the primary task of hate speech detection. This enables the model to conduct efficient and precise sentiment analysis, substantially improving the performance of the primary task.

## Appendix C: Experiment Setup

### C.1 Dataset

Considering the practical challenges in obtaining a large amount of data generated by LLMs, we conduct experiments using widely used public hate speech datasets. First, we utilize a sentiment dataset and two publicly available hate speech datasets. The hate speech datasets mimic the potential output generated by an LLM-based web application. Moreover, we select two other public hate speech datasets to assess out-of-distribution performance. Lastly, we incorporate a hate speech dataset generated by GPT-3 to evaluate detection performance, specifically targeting the outputs of LLM-based web applications. We have carefully ensured that there is no overlap between the datasets used in our experiments. Our proposed method is inherently language-agnostic and can be easily extended to support multilingual hate speech detection. However, to expedite evaluations, we use only English data.

*C.1.1 Hate Speech Datasets.* We utilize two public hate speech datasets, i.e., SBIC [27] and HateXplain [21]. We select these datasets to simulate outputs from an LLM-based web application under evaluation, as the text in these datasets is collected from the Internet, resembling the samples that are scraped from the Internet and used for training an LLM. For both datasets, we randomly choose 16 to 1024 samples for few-shot learning.

- **SBIC** [27]. We utilize the Social Bias Inference Corpus (SBIC) to create one of the few-shot hate speech detection datasets. The corpus comprises structured annotations of 150,000 social media posts (e.g., Reddit or Twitter) spanning approximately 1,000 demographic groups. In line with previous studies defining hate speech [2, 6], we employ annotations for offensiveness and demographic group targeting to determine whether a post qualifies as hate speech, i.e., if a post uses language targeting offensively a demographic group.
- **HateXplain** [21]. We also leverage the HateXplain dataset to create a few-shot hate speech detection dataset. This dataset consists of 20,000 samples gathered from Twitter and Gab, with labels categorized into three groups: Hate speech, offensive language, and normal language. Each sample is paired with a relevant target group and explainable rationales.

*C.1.2 Sentiment Datasets.* We adopt the Twitter for Sentiment Analysis (T4SA) [30] to assemble the training and testing sets for the

intermediate task of Sentiment-Aided Hate Speech Detection. This corpus accumulates Twitter posts over a span of six months, beginning in December 2016. It determines the text sentiment polarity (negative = 0, neutral = 1, positive = 2) for each post using a combined LSTM-SVM structure. The dataset contains approximately 1.18 million samples. We randomize and divide the dataset in a 99:1 ratio for the training and testing of the intermediate task.

*C.1.3 Out-of-Distribution Datasets.* To gauge the out-of-distribution performance of our approach, we employ the following two datasets for evaluation:

- **HS18** [7]. This corpus was collected from a white supremacist forum called Stormfront. Each sample in this dataset has a binary label indicating whether it is hate speech.
- **Ethos** [23]. This corpus comprises comments from social media platforms, i.e., YouTube and Reddit. Each comment has a binary label designating it as hate speech or not.

*C.1.4 Hate Speech Datasets for LLM Outputs.* To validate the performance of our approach in detecting hate speech generated by LLMs, we use the **ToxiGen** dataset [14]. ToxiGen is a large-scale hate speech dataset generated using GPT-3, which includes 274,000 toxic and benign statements and involves approximately 13 minority groups. The ToxiGen dataset contains a significant amount of implicit hate speech, which further challenges the capabilities of our hate speech detection method.

### C.2 Performance Metrics

Following ToKen [2], we use the macro F1-score to evaluate various classifiers for hate speech detection. The macro F1-score is a widely used metric in NLP tasks. It measures the harmonic mean of precision and recall. The macro F1-score for a multi-class classification problem is determined by averaging the individual F1-scores for each class. The macro F1-score for a specific class $i$ is given by

$$\text{F1}_i = \frac{2 \cdot \text{precision}_i \cdot \text{recall}_i}{\text{precision}_i + \text{recall}_i}. \tag{9}$$

The macro F1-score across all classes is given by

$$\text{Macro F1-score} = \frac{1}{N} \sum_{i=1}^{N} \text{F1}_i, \tag{10}$$

where $N$ is the number of classes. The macro F1-score provides a balanced performance assessment by giving equal importance to each class, regardless of its size.

### C.3 State-of-the-Art Methods

Although there are various works based on traditional machine learning (e.g, the SVM with n-grams [22]) or deep learning (e.g., the standard neural network with multi-head attention [34]), state-of-the-art (SOTA) hate speech detection solutions are mostly based on pretrained transform based LMs, such as BERT, BART, or GPT. Therefore, the following SOTA methods are considered as benchmarks to be compared with the proposed method.

- **BERT** [9]: This method fine-tunes the BERT-base-uncased model and appends a fully connected layer as the classification head.

- **BART-P** [18]: This approach incorporates prompts with manually constructed templates and harnesses the pre-trained BART-Large model. For training on the SBIC dataset, the approach adopts the strategy presented in ToKen [2] to use the template *"Hate speech?"*. For the HateXplain dataset, it employs the template *"Offensive towards a group?"*.
- **ToKen** [2]: This method is anchored on the BART-Large model and integrates the principles of task decomposition and knowledge infusion.
- **HARE** [31]: This method is built upon the reasoning capability of LLMs, which are fine-tuned using chain-of-thought (CoT) prompts to train the model in identifying text categorized as hate speech. We employ the state-of-the-art T5-large model and Fr-HARE method, which demonstrated the best performance in [31].
- **BERT-H** [21]: This method leverages the annotated labels in the HateXplain dataset to discern which segments of the posts should be emphasized. It integrates an attention supervision mechanism into the BERT model targeting those segments. It aligns the attention scores of the CLS token in the final layer with genuine attention. It also supplements the attention loss to the label prediction loss for loss computation.
- **BERT-M** [16]: This approach initially fine-tunes the Masked Rationale Prediction (MRP) intermediate task before subsequently fine-tuning for hate speech detection.

## C.4 Hyperparameter Settings

This section provides an overview of the training process and hyperparameters used for each dataset.

To construct the few-shot training sets, we adhere to the methodology outlined by ToKen [2], employing stratified sampling from the SBIC and HateXplain corpora. We draw samples from the inoffensive (termed "normal" in HateXplain), offensive but not targeting any specific group (labeled "offensive" in HateXplain), and hate speech categories in a ratio of 1:1:2. We create datasets of varying sizes, ranging from 16 samples to 1024 samples. Given the inherent instability of few-shot learning, we compile ten distinct datasets with the size of $N$ per dataset, using ten different random seeds. We then compute the average performance under each set size.

The fine-tuning of models in our approach utilizes a batch size of 16, and the fine-tuned models incorporate early stopping with a maximum of 20 epochs. The learning rate is set to $1 \times e^{-5}$ for the SBIC, HateXplain, and T4SA datasets. Our experiments are conducted using PyTorch 2.0 on an NVIDIA A100 GPU. The AdamW optimizer is employed, complemented by a linear warm-up schedule and a weight decay of 0.01.

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
