# OpenReview forum: "SAHSD: Enhancing Hate Speech Detection in LLM-Powered Web Applications via Sentiment Analysis and Few-Shot Learning"
_ACM.org/TheWebConf/2025/Conference — WWW 2025 Poster_

### Official Review · Reviewer_ZmtG · 2024-11-18

**Novelty:** 2
**Technical Quality:** 4

**Review:**

Summary:

The rise of large language model (LLM)-powered web applications has amplified concerns about moderating hate speech due to vulnerabilities in training data. Existing methods face challenges with limited annotated data, especially for real-time moderation. This paper introduces Sentiment-Aided Hate Speech Detection (SAHSD), a novel approach that treats hate speech detection as a few-shot learning task. SAHSD leverages sentiment analysis to fine-tune pre-trained language models, enhancing detection accuracy even with minimal training samples. By integrating sentiment and hate speech prompts, SAHSD achieves remarkable performance, including an F1-score of 0.99 with just 64 training samples. It outperforms advanced methods like ToKen and GPT-4, showing significant improvements on datasets like SBIC and HateXplain. These results highlight SAHSD’s potential to improve content moderation on LLM-driven platforms, promoting a safer and more inclusive web environment.

Pros:

1. The experimental data results in the article are very detailed.
2. The illustrations in the article are well-designed and easy to understand.

Cons:

1. The authors may have used an incorrect ACM template, omitting line numbers. This likely stems from missing the "review" keyword in the conference-recommended `documentclass`. This oversight could cause difficulties for reviewers.
2. The main part of the article lacks a *Related Work* section, which is an essential and indispensable component of a research paper. It introduces prior similar work and should not be omitted or relegated to the appendix.
3. The authors use *sentiment analysis as an intermediate task* to address annotator biases in existing datasets. However, sentiment analysis results can also be influenced by prior annotator biases. Furthermore, the final hate speech judgment relies on publicly available annotated datasets, which inherently include human biases. The authors did not adequately discuss this issue in the paper.
4. In the experiments, the authors specify the LLaMA2 parameter size as 13B, but LLaMA2 also has a 7B version. The difference between 0.34B and 7B is not very significant (7B can also run on normal computers), and the authors did not clearly explain the benefits of lower computational costs in this context. They should consider comparing their approach with similar works rather than only including BERT (as shown in Table 1).
5. Generally, hate speech detection tasks are performed by service providers or community operators who possess ample computational and data resources. However, the article claims that this strategy is particularly suited for low-resource settings and achieves a high F1 score with just 64 samples. This conflicts with real-world scenarios, and the authors should describe the specific use cases in more detail.
6. The authors did not discuss whether the excellent performance achieved might result from overfitting.
7. Figures 2 and 3 are of the same type and should be combined into a single figure. If formatting is an issue, the authors could use a cross-column layout.
8. The authors did not analyze why there was a significant performance improvement with sample sizes of 16-64 in Figures 2 and 3 but merely summarized this observation.
9. Model parameters influence experimental results, and the authors should list the parameter details of the compared models.
10. The authors did not explain why, in Figure 4, the F1 scores of other LLMs do not vary with changes in sample size.
11. The T4SA dataset is mentioned in the appendix but not in the main part of the paper.
12. The authors should improve the formatting of the paper, as it currently leaves much room for improvement.

**Questions:**

1. Could the outstanding performance of SAHSD be attributed to overfitting in small-parameter models? Can this be proven or disproven?
2. Are the *low-resource settings* mentioned in the paper meaningful in real-world scenarios? Projects implementing hate speech detection often have abundant resources.
3. Why is the T4SA dataset not mentioned in the main part of the paper?

**Reviewer Confidence:**

3: The reviewer is confident but not certain that the evaluation is correct

**Scope:**

2: The connection to the Web is incidental, e.g., use of Web data or API

---

### Official Review · Reviewer_PZh2 · 2024-11-24

**Novelty:** 4
**Technical Quality:** 5

**Review:**

Strengths:
1) SAHSD introduces a novel methodology by incorporating sentiment analysis into hate speech
detection. By leveraging sentiment patterns, particularly negative sentiment, the model enhances hate
speech detection even with limited annotated data. This approach is particularly beneficial in hate
speech detection, where collecting sensitive data can be challenging.
2) SAHSD demonstrates robust performance in few-shot learning scenarios, effectively detecting hate
speech with a small number of samples. This characteristic is valuable when annotated data is
scarce, enabling effective hate speech detection with minimal data. The approach is also resource-
efficient, making it advantageous in terms of cost and computational requirements.
3) SAHSD outperforms large language models like GPT-4 in specific tasks. This suggests that
SAHSD is not only effective but also more efficient, offering better performance for hate speech
detection without the need for extensive computational resources.
Weaknesses
1) The SAHSD model enhances hate speech detection performance by incorporating sentiment
analysis. However, not all hate speech directly correlates with negative sentiment. For instance,
sarcasm, which may carry a neutral sentiment, can also constitute hate speech. Similarly, some hate
speech may convey an exaggerated positive sentiment (i.e., extreme nationalism). Therefore, the
assumption that negative sentiment consistently aligns with hate speech could limit the model&#39;s
effectiveness.
2) While SAHSD focuses on textual sentiment and specific words, it may fail to capture essential
contextual elements necessary for accurate hate speech detection. Hate speech often gains its
meaning from prior statements or conversational context, making it challenging to detect hate within
isolated statements. This paper does not discuss contextual, conversation-based hate speech
detection, which could lead to misclassification when the model relies solely on individual statements
without broader context.
3) The use of sentiment analysis and prompt engineering in generating hate speech detection data
could pose ethical risks if the techniques are misused. For instance, an understanding of sentiment
and hate speech markers could, in theory, be exploited to create more sophisticated and undetectable
hate speech. Although this is not directly a limitation of SAHSD’s technical approach, the ethical
considerations of dual-use technology should be acknowledged and managed carefully.

**Questions:**

(see the review above)

**Reviewer Confidence:**

4: The reviewer is certain that the evaluation is correct and very familiar with the relevant literature

**Scope:**

3: The work is somewhat relevant to the Web and to the track, and is of narrow interest to a sub-community

---

### Official Review · Reviewer_LbKX · 2024-11-25

**Novelty:** 4
**Technical Quality:** 3

**Review:**

This paper proposes a novel method (SAHSD) to enhance the performance of hate speech detection given limited training samples of hate speech. Hate speech detection has always been an important task for researchers and platform moderators, as these texts can pollute the online environment and even incite real-world violence. The authors employ various techniques, including few-shot learning and prompt tuning, to contribute a capable hate speech detector. They also present the paper with high clarity.

However, I have the following concerns related to motivation, methodology, and evaluation. I think the biggest concern from my side is the motivation, which I find hardly convincing.

- **Motivation**

There are many publicly available datasets of annotated hate speech examples, such as HateEval [a], DynaHate [b], MHS [c], and others (CMS, HateX, HTPO, etc.). The community already has sufficient data for training a generalizable hate speech detector. For example, the MHS dataset alone consists of 50K data points with fine-grained annotations (e.g., disrespect, attacking, genocide, etc.). In this context, it is difficult to convince reviewers that the proposed method's motivation is based on the limited training datasets.

- **Intuition of the Proposed Methodology**

Sentiment classification and hate speech classification are two completely different tasks, although they might be positively correlated. A sentence expressing positive sentiment does not necessarily result in hate speech, and vice versa. For example:  ``*This man is a true hero for protecting our jobs from immigrants.”* The sentiment is positive, but it reveals implicit hateful ideology, i.e., discrimination against immigrants. In such cases, will SAHSD provide the correct prediction, given that sentiment analysis may offer misleading hints?

From this perspective, I believe the real challenge in hate speech detection lies in capturing the subtle, implicit, and variable forms of hate in seemingly harmless texts. Sentiment-aided detection may struggle to assist with this task, as it is more suited for identifying explicit hate in texts.

- **Evaluation**

The authors evaluate the performance of SAHSD and compare it with several baselines. One of the most important baselines is supervised fine-tuning, which is a common practice for tasks such as hate speech detection and toxic language detection. For supervised fine-tuning, it is less reasonable to strictly limit the sample size (from 16 to 1024) of the training dataset, as this training paradigm is designed to learn from extensive data samples to differentiate between hate and non-hate speech.

I understand the motivation behind this setup is that the authors are concerned about limited datasets and computing resources. However, these issues are less significant in the community. As I previously mentioned, there are multiple publicly available datasets, and training a BERT or even a larger model on 10K samples requires no more than several GPU hours.

Additionally, public APIs or models for hate speech detection, such as Google’s Perspective API (identity attack), Detoxify, LFTW, TweetHate, and others, are available. These existing state-of-the-art detectors should also be included in the baselines to demonstrate the superior effectiveness of the proposed method.

----------------------
[a] Basile, Valerio, et al. "Semeval-2019 task 5: Multilingual detection of hate speech against immigrants and women in twitter." *Proceedings of the 13th international workshop on semantic evaluation*. 2019.

[b] Vidgen, Bertie, et al. "Learning from the worst: Dynamically generated datasets to improve online hate detection." *arXiv preprint arXiv:2012.15761* (2020).

[c] Sachdeva, Pratik, et al. "The measuring hate speech corpus: Leveraging rasch measurement theory for data perspectivism." *Proceedings of the 1st Workshop on Perspectivist Approaches to NLP@ LREC2022*. 2022.

**Questions:**

1. **Motivation**: Given the availability of large, publicly annotated hate speech datasets (e.g., MHS, HateEval), how is the limitation of training data a valid motivation for the proposed method?
2. **Methodology**: How does the sentiment-aided approach handle the challenge of detecting subtle, implicit hate speech, where sentiment analysis shows a positive emotion?
3. **Evaluation**: Why are the training sample sizes for supervised fine-tuning strictly limited, and why are state-of-the-art public APIs or models for hate speech detection not included in the baseline comparisons?

**Reviewer Confidence:**

4: The reviewer is certain that the evaluation is correct and very familiar with the relevant literature

**Scope:**

4: The work is relevant to the Web and to the track, and is of broad interest to the community

---

### Official Review · Reviewer_URwZ · 2024-12-01

**Novelty:** 5
**Technical Quality:** 5

**Review:**

Dependency on Sentiment Datasets: While effective with few samples, the model's reliance on sentiment datasets may limit its applicability in contexts where such data is unavailable.
Potential Bias: The model's performance may still be influenced by biases in the training data, raising concerns about its reliability in real-world applications.
Performance Dependence on Sample Size: Although designed to address limited data for hate speech, the model's performance improves significantly with more samples, suggesting its architecture may be less optimal for truly low-data scenarios.

**Strength**
- SAHSD delivers exceptional results, achieving an F1-score of 0.99 with only 64 training samples, demonstrating its effectiveness in low-data scenarios.
- The model exhibits strong generalization, outperforming existing models like GPT-4 in zero-shot scenarios, making it highly valuable for real-world applications.

**Weakness**
- (W1) Dependency on Sentiment Datasets: While effective with few samples, the model's reliance on sentiment datasets may limit its applicability in contexts where such data is unavailable.
- (W2) Potential Bias: The model's performance may still be influenced by biases in the training data, raising concerns about its reliability in real-world applications.
- (W3) Performance Dependence on Sample Size: Although designed to address limited data for hate speech, the model's performance improves significantly with more samples, suggesting its architecture may be less optimal for truly low-data scenarios.

**Questions:**

Please address W1-W3 comments.

**Reviewer Confidence:**

1: The reviewer's evaluation is an educated guess

**Scope:**

3: The work is somewhat relevant to the Web and to the track, and is of narrow interest to a sub-community

---

### Official Review · Reviewer_jDFK · 2024-12-02

**Novelty:** 5
**Technical Quality:** 6

**Review:**

Sentiment-aided hate Speech Detection (SAHSD) is an approach designed to detect hate speech in web applications. The sentiment analysis portion of the research is very interesting and difficult to identify.

**Questions:**

None

**Reviewer Confidence:**

1: The reviewer's evaluation is an educated guess

**Scope:**

4: The work is relevant to the Web and to the track, and is of broad interest to the community